# Why do people sell their kidneys? A thematic synthesis of qualitative evidence

**Bijaya Shrestha** [1]*, **Luechai Sringernyuang** [2], **Manash Shrestha**[3], **Binita Shrestha**[4], **Anuska Adhikari** [1], **Dev Ram Sunuwar**[5], **Shiva Raj Mishra**[6], **Bipin Adhikari**[7]

**1** Center for Research on Education, Health and Social Science, Kathmandu, Nepal, **2** Contemplative Education Center, Mahidol University, Nakhon Pathom, Thailand, **3** Asia Pacific Malaria Elimination Network (APMEN) Vivax Working Group, Kathmandu, Nepal, **4** Independent Researcher, Kathmandu, Nepal, **5** Department of Nutrition and Dietetics, Armed Police Force Hospital, Kathmandu, Nepal, **6** Melbourne School of Population and Global Health, University of Melbourne, Parkville, Australia, **7** Mahidol-Oxford Tropical Medicine Research Unit, Faculty of Tropical Medicine, Mahidol University, Bangkok, Thailand

* bijayashresthanepal@gmail.com

**Data Availability Statement:** All relevant data are contained within the manuscript.

**Funding:** The authors received no specific funding for this work.

## Abstract

Globally, demands for the kidneys have surpassed supply both living and deceased donors. High demands relative to the availability have made the kidney one of the most saleable human organs. The main objective was to explore the drivers of kidney selling. Literature related to kidney selling and its drivers was explored in three databases including MEDLINE (PubMed), Scopus (Elsevier), and JSTOR covering the period from 1987 to 2022. A total of 15 articles were selected, which underwent thematic analysis. Investigators independently assessed the articles for relevance and study quality to synthesize the data. The thematic analysis involved a critical approach to understanding the reasons for kidney selling by examining power disparities and social inequities. Kidney selling and the underlying reasons for it showed similarities across various geographic regions. Several factors were identified which increased individuals' vulnerability for kidney selling. At the micro level, poverty and illiteracy emerged as significant factors. Lack of financial safety nets obliged family to resort to kidney selling which helped to alleviate poverty, resolve debt, and other urgent financial issues. Nonetheless, the revenues from kidney selling were also used to purchase luxury items (diverting away from investing in livelihood expenses) such as buying motorbikes, mobile phones and televisions. Family, and gender responsibilities also played roles in kidney selling such as obligations related to paying dowry made parents particularly vulnerable. Surprisingly, a few victims of kidney selling later adopted kidney brokering role to support their livelihood. Kidney selling was further fostered by lack of stringent policy to regulate and monitor background checks for kidney transplantation. There were myriad factors that affected individual's vulnerability to kidney selling which stemmed from micro (poverty, illiteracy), meso (weak legal system, lacking stringent institutional policy, regulatory framework) and macro (social inequalities, corruption, organ shortage, insufficient health infrastructure) levels.

**Competing interests:** The authors have declared that no competing interests exist.

## Introduction

Kidney transplantation is widely recognized as the preferred treatment for end-stage renal diseases worldwide [1]. This procedure takes place in over a hundred different countries. However, the supply of kidneys has not been able to meet the growing demand. Kidneys are in high demand globally, with low- and middle-income countries (LMICs) being major suppliers. Poverty, vulnerability, and lack of awareness have been identified as drivers that facilitate kidney selling [2–4]. However, the selling of kidneys is not simply a transactional act; it often serves as a means of social support to alleviate economic burdens and is driven by short-sighted impulses influenced by suggestions from peers and relatives [5]. Kidney selling could be solely initiated by the kidney sellers to an institution or a person without necessarily involving a broker. Kidney trafficking, on the other hand, can involve acts of exploitation, coercion, and inducement to sell to an institution or a person, often with the involvement of a broker [6]. Kidney sellers are often unaware about the consequences of selling kidneys such as disabilities of various extent including weaknesses, chronic kidney disease, stigma and unforeseen complications [7].

To meet the outweighing demands of kidney, the supplies often follow both legal and illegal mechanisms [3]. The primary reason for the demand has been attributed to chronic kidney diseases. Individuals from high-income countries with long waiting lists for kidney transplants may travel to countries with less stringent regulations or where organ trafficking occurs to access organs more quickly [8]. The supply of kidneys from deceased donors is limited by factors such as consent processes, medical suitability, and logistics. Living donation is tightly regulated to ensure the well-being and informed consent of the donor [9]. Unfortunately, illegal organ trafficking has been a common occurrence in LMICs which exploits vulnerable individuals facing desperate living conditions [3]. Although organ trafficking has been condemned and deemed illegal, it remains persistent because of the lack of regulatory measures [10]. The legal framework guiding kidney selling varies across countries and are dependent on the law (policy) itself and its implementation. The policies guiding kidney selling are in general deemed to regulate organ donation and thus prevent exploitation, protect vulnerable individuals, and ensure the ethical and equitable allocation of organs for transplantation [11]. The policies also guide and safeguard the terms and conditions for living organ donations, delineating the acts of voluntary and altruistic donations. In some countries, there are well regulated organ donor programs that allow kidney transplantation from living donors [7]. The legal framework for the kidney trafficking is covered under the human trafficking binding the concerns related to prosecution, protection and prevention [6].

The majority of kidney transplantation in LMICs are contributed by trafficking [4, 12]. Transplantation, trafficking and trade all run in parallel, making it hard to differentiate the associated activities. In recent years, kidney transplant technology has advanced significantly, improving its success rates [13]. The surgical techniques used in kidney transplantation have become more refined and minimally invasive. Increasingly, modern medical technologies have aided smooth transplantation thus ironically promoting the kidney transplantation [14]. Nonetheless, ethical arguments related to the kidney selling remains unaddressed by legal frameworks. Even if legal frameworks are meant to safeguard against the exploitation involved in kidney selling, the act of selling kidneys among the vulnerable poor compromises the central tenet of autonomy in the consent process. The consent itself for vulnerable donors is affected by undue inducement through incentives involved in the transaction. Simultaneously, recipients are vulnerable because their opportunity for life will be compromised if they do not accept donated organs from anyone, including vulnerable donors. For both donors and recipients, the nature and degree of vulnerability tend to cloud the ethical principles of the consent

process. In addition, allowing the selling of organs from mostly poor and vulnerable donors is likely to promote inequity, as most buyers are privileged and can afford organs, while the poor with need for kidney transplantation may face inability to purchase an organ. Viewed from the perspective of the basic right to health, buyers have moral ground to afford organs for their well-being, nonetheless, this tends to support those who are privileged, leaving the poor in need of organs further vulnerable [15].

The medical consequences of kidney selling, both for the sellers (donors) and the recipients are different. In case of donor, kidney donation surgery carries risks inherent in any major surgical procedure, such as bleeding, infection, blood clots, and anesthesia-related complications [16]. Donors may further experience psychological and emotional challenges related to the donation process, including stress, guilt, and regret [5]. In case of recipients, despite advances in immunosuppressive medications, there is always a risk of rejection of the transplanted kidney. Transplant recipients may experience long-term complications, including cardiovascular disease, diabetes, bone disease, and an increased risk of certain cancers [17]. In illegal kidney trade in unregulated markets, there is a lack of proper medical oversight, which increases the chances of medical errors, poor surgical outcomes, and inadequate post-operative care [18]. In the compromised setting of kidney transplantation, there is a risk of acquiring HIV and Hepatitis due to potential exposure to contaminated blood or inadequate screening measures [19].

The main objective of this systematic review was to explore the reasons for kidney selling around the globe. Kidney selling is a global phenomenon, nonetheless there is a paucity on how and under what circumstances kidney selling occurs. In this review, we use a critical medical anthropological approach to explore social inequity and power differentials as the primary contributors of health and health care. Social inequalities, as viewed through the lens of Critical Medical Anthropology (CMA), encompass a consideration of the broader social, economic, and political structures that influence health outcomes and experiences [20]. Within CMA, social inequalities are defined as the unequal distribution of resources, opportunities, and power within society, which profoundly impact health and well-being. These inequalities are not merely the result of individual choices or biological differences but are deeply rooted in systemic injustices such as racism, sexism, classism, and other forms of oppression. Similarly, in CMA, power differentials refer to the uneven distribution of power within society and how these dynamics shape health experiences, outcomes, and access to healthcare. This review also analyzes the relationship between health status at an individual and societal level [21].

## Materials and methods

We followed the guidelines of ENTREQ (Enhancing Transparency in Reporting the Synthesis of Qualitative Research) for this qualitative systematic review [22]. The protocol was registered in the PROSPERO database (CRD42020197541).

### Search strategy

We conducted a systematic literature search in the electronic databases of MEDLINE (PubMed), Scopus (Elsevier), and JSTOR. The search strategy included the keywords "kidney" [AND] "sale" [OR] "trade" [OR] "deal" [OR] "Commerce" (S1 Table).

We also manually searched the reference lists of published reviews and used Google Scholar to identify additional studies that may have been missed in our electronic database search. The additional papers were also searched after following the references from the selected articles. Since the World Health Organization declared organ sales an illegal act in 1987 [23], we searched for studies published after 1987 and up to December 2022.

## Study selection

All studies were included that offered qualitative evidence on selling kidneys for commercial reasons including motivations to sell their kidneys. Peer-reviewed qualitative research utilizing face-to-face interviews, focus group discussions, ethnography, case studies, and comparative content analysis were included. Only studies published in English language were included in this review. In the final selection, even though we found a substantial discussion about the phenomenon of kidney selling in a few quantitative studies [24–26], we did not include them in the review as these studies did not elicit novel insights on how the factors contributed to kidney selling. We also excluded non-original research such as editorials, reviews, commentary, essays, arguments, debates, news, opinions, and reports.

EndNoteTM X9 (Clarivate Analytics, Philadelphia, PA) was used to import the records into a database. Two reviewers (BS and MS) independently screened the study titles and abstracts retrieved from the search and discarded the studies that did not meet the inclusion criteria. Duplicate papers were removed (n = 61), first with the help of EndNote and later by a manual check. The full texts of the remaining studies were retained and subsequently reviewed for inclusion.

## Data extraction and quality assessment

Data were extracted from the selected articles using a pre-specified extraction table based on thematic synthesis for qualitative study [27]. For each selected article, we recorded the first author, year of publication, study country, duration of data collection, and method of data collection (S2 Table). To assess transparency of the reporting of each selected study, we utilized the Consolidated Criteria for Reporting Qualitative Research (COREQ), a 32-item checklist designed to promote comprehensive reporting of qualitative studies. The COREQ framework includes several important aspects related to research team and reflexivity, study design, analysis, and findings. The guidelines place a significant emphasis on the requirement to report details regarding the research team who participated in the study, including their backgrounds, areas of specialization, and any potential biases or conflicts of interest. Reflexivity is the ability of the researcher to see and accept their own impact on the research process. COREQ encourages researchers to describe their reflexivity by addressing their preconceptions, assumptions, and potential impact on data collection and analysis. On the research design; it entails outlining the research topic, the study's goals and objectives, as well as the setting in which it was accomplished. The use of sampling techniques, participant recruitment, and any ethical considerations are also reported by the checklist. Transparency in reporting the data analysis process is also emphasized along with the techniques utilized for data coding, categorization, and topic development [28] (S3 Table).

## Data synthesis and analysis

The thematic analysis took an approach outlined by Critical Medical Anthropology (CMA) to understand the reasons for kidney selling by examining power disparities and social inequities. The power disparities and social inequities among the rich and poor in societies are governed by wealth, knowledge, geographic locations, citizenship, ethnicities, gender, class and forms of stratifications. As social complexity increases, inequity tends to increase along with a widening gap between the poorest and the wealthiest members of the society [29]. We analyzed the situation at different levels, starting from the individual level to the macro-social level. Using a CMA theoretical lens, we delved deeper into the experience of the kidney sellers and understand their decisions in light of power dynamics and the influence of dominant socio-cultural and economic forces. The CMA approach also highlights the importance of political and economic forces, including the exercise of power, in shaping health, disease, illness experience,

and health care, which supplements the culturally sensitive analysis of human behavior grounded in anthropological methods. It critically examines the ways in which power, inequity, and social structures shape health outcomes and the experiences of individuals and communities. Furthermore, it examines the social origin of diseases by analyzing the policy, resources allocation in terms of health and health care [21].

Participant quotations were extracted verbatim from the selected studies into MAXQDA 2020 (VERBI Software, 2019). Two reviewers (BS and MS) separately performed line-by-line coding and identified initial descriptive themes. List of themes from the analysis were refined and finalized after discussing with the third reviewer (BA). All of the authors agreed on the final themes, which are the basis of the results in this review.

## Results

### Study selection

A total of 2,030 articles were generated from the electronic search (Fig 1). A total of 1349 articles were excluded in the initial screening. The exclusion of articles was carried out by carefully reviewing the titles and assessing their relevance to the research question, followed by a thorough examination of the content. The clinical articles and pre-prints article were excluded in the initial screening. In addition, 681 articles were further screened of which 61 were duplicates, and thus were excluded. Out of 620 articles, 332 articles such as archive, review, non-original article, essay, including those which lacked full text were excluded. After screening for the relevance of the topic and study objectives from the title and abstracts, 288 articles were retained for full-text review. Assessment of the full text against the inclusion criteria resulted in the selection of 15 studies with qualitative evidence of commercial kidney donors' reasons for undergoing nephrectomy.

The studies selected for final analysis showed considerable geographic variability: Asia-Pacific (India, Pakistan, Bangladesh, The Philippines, Indonesia, Nepal), Europe (Moldova, The Netherlands), Middle East (Israel, Iran), North America (Canada), South America (Brazil), and Africa (South Africa) (Fig 2).

Major themes from the articles in this review are displayed in the figure as speech bubbles and are detailed below:

1. Economic conditions: Poverty was identified as the primary reason for kidney selling among individuals from the Indian Subcontinent. Economic hardships and the need to alleviate poverty were consistently observed as drivers for engaging in kidney selling.

2. Family and societal responsibilities: The articles highlighted the influence of family and societal norms on kidney selling practices. Fulfilling family demands and adhering to societal expectations were additional factors contributing to the decision to sell a kidney.

3. Desperation to alleviate poverty: In some cases, families were driven to sell kidneys out of desperation to fulfil debt traps, and social obligations. The articles shed light on the dire circumstances faced by these individuals and the extreme measures taken to improve their economic situation.

4. Gender roles: Gender roles within families played a role in kidney selling. Males sometimes resorted to selling their organs to acquire money for dowry, while females sold their organs due to the physical labor expected of men in their families.

5. To purchase material goods: Although poverty was the primary driver, the articles also highlighted instances where individuals sold their kidneys to acquire luxury items such as motorbikes, phones, or to invest in additional land deterring away from supporting livelihood expenses.

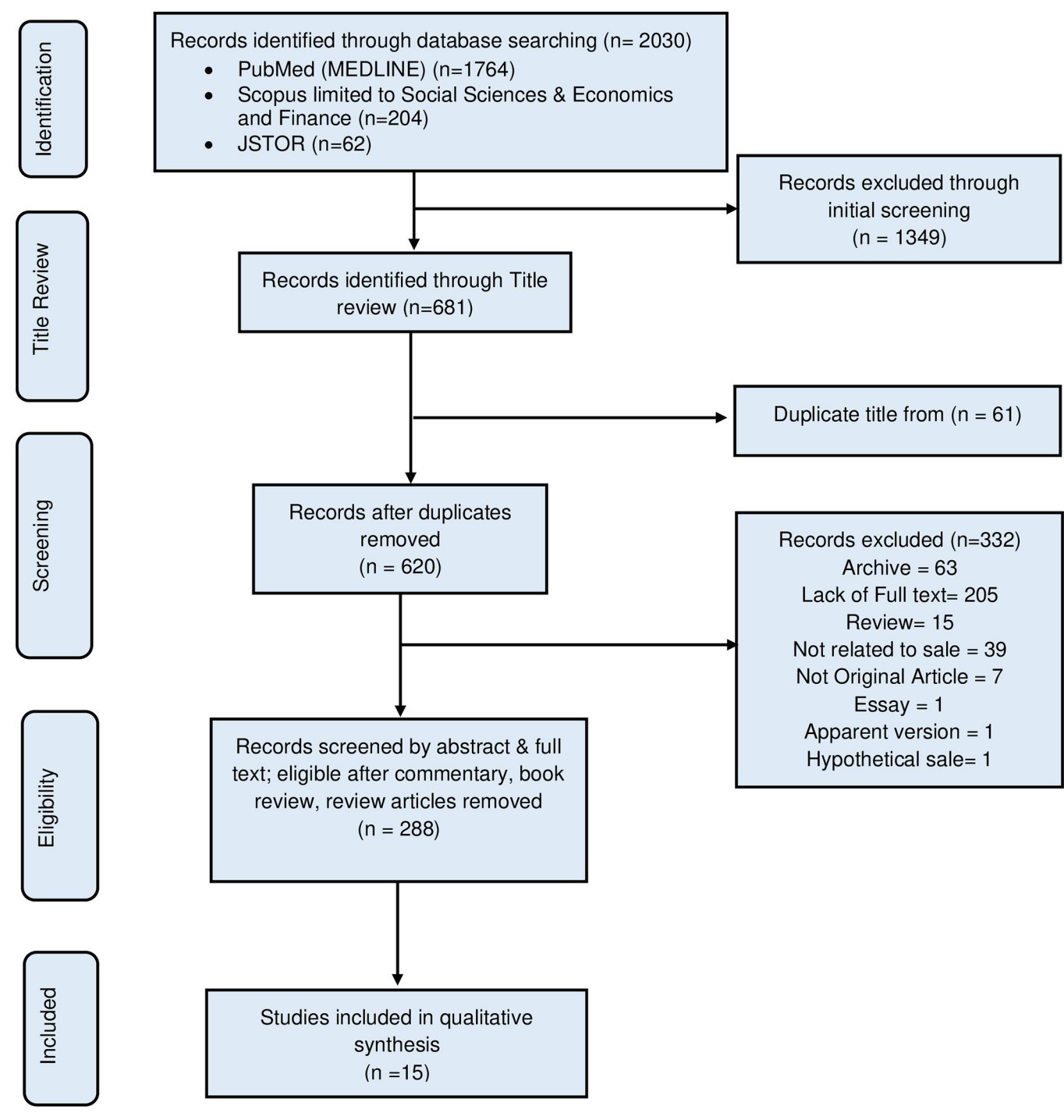

**Fig 1. The PRISMA flowchart showing the inclusion and screening in the review.**

6. Role of brokers: The presence of brokers facilitating kidney selling was noted in countries such as Bangladesh, Nepal, India, and the Philippines. These intermediaries played a significant role in connecting potential sellers with potential buyers.

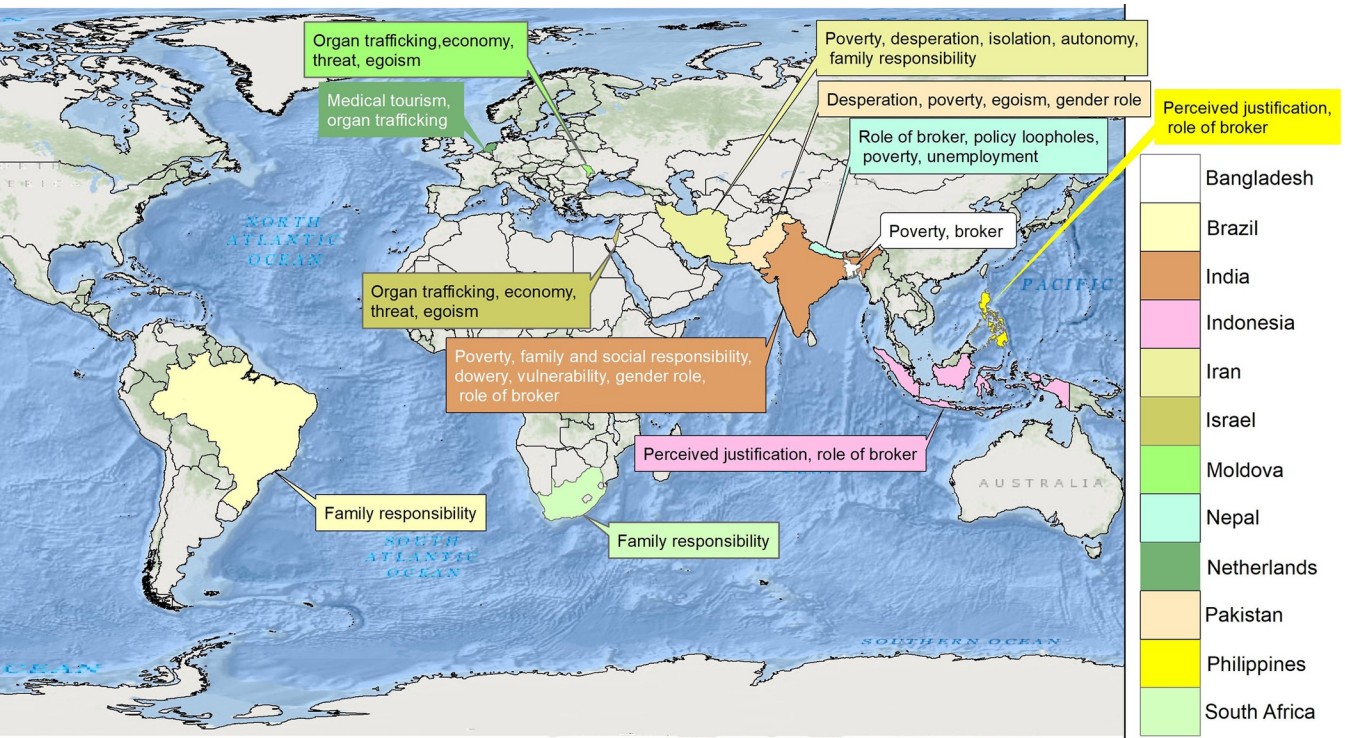

**Fig 2. Geographic coverage and themes from the places.** Image was created by the authors in ESRI's ArcGIS 10.3.1; no copyrighted material is used in this image. World Imagery ESRI Tile Layer was used as basemap located in: https://services.arcgisonline.com/ArcGIS/rest/services/World_Imagery/MapServer. Service Layer Credits: Source: Esri, Maxar, GeoEye, Earthstar Geographics, CNES/Airbus DS, USDA, USGS, AeroGRID, IGN, and the GIS User Community.

7. Inadequate policies and transnationalism: The articles emphasized the existence of lax policies that were unable to effectively address the issue of transnational kidney trafficking. Particularly in countries with open borders, the shortcomings of policies in addressing the challenges of kidney selling and cross-border organ trade were evident.

These themes offer insights into the complex factors driving kidney selling and highlight the need for comprehensive policies and interventions to address this issue effectively. Despite the reference of Canada, no themes were derived from it and it was only referred in comparison to Bangladesh [30].

## Comprehensiveness of reporting

The comprehensiveness of reporting varied across the studies, with the number of reported details ranging from 6 to 25 of the 32 items required by the COREQ (S2 Table). Five out of fifteen studies provided details on less than ten of the 32 items [7, 30–33]. Eight studies reported the personal characteristics but there was no information about the relationship building of the researchers with the participants [2, 3, 5, 34–37]. Only one reported relationship building with the participants [38]. One study that reported policy-level content analysis was not applicable to be assessed on the items of personal characteristics of the researchers, relationship building, and data collection. The majority of the studies described the themes, quotations, consistent data findings and data collection process but there were few who described in details. Only three articles described the reasons for non-participation [5, 36, 39], while only two stated when and where the field notes were taken [5, 40]. Meanwhile, only one explained

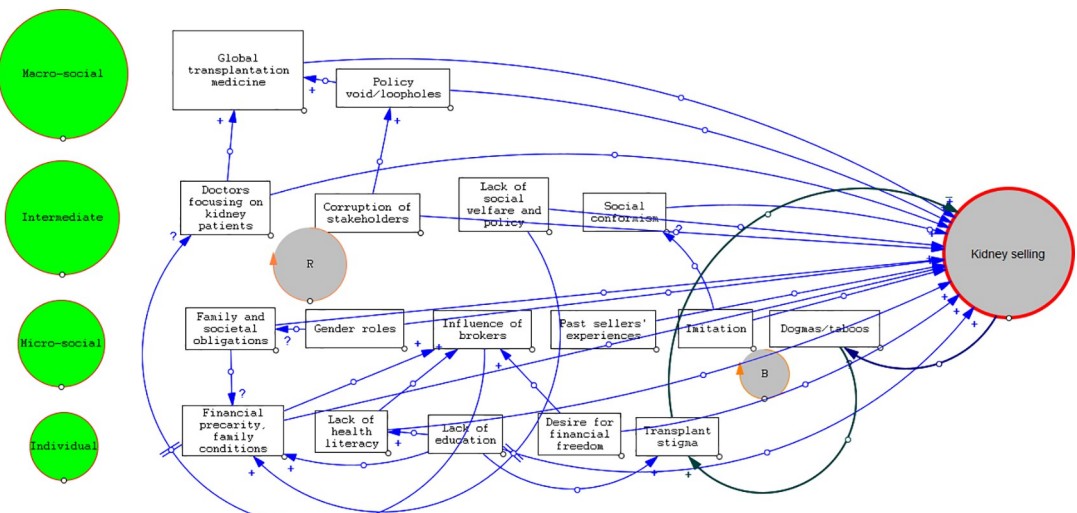

**Fig 3. Factors influencing kidney selling in Nepal.** This figure was developed by authors based on the findings of this review and describes a web of multi-layered factors operating at the micro, meso and macro level. (+) sign denotes a positive link, and (−) sign denotes a negative link where a change in influencing factor is in the same direction as the influenced element. (?) sign denotes unclear direction of relationship between two causal pairs. '‖'indicates there is a delay where a change in influencing factor produces change in influenced element only after an interval of time. 'R' denotes a reinforcing loop and 'B' denotes balancing loop. To sum up, this figure shows the complexity in interaction between factors operating at the individual, micro-social, intermediate and macro level (green circles on left). For example, financial precarity (bottom left) is contributed by a lack of social welfare system and not having basic education for livelihood. Combination of these factors highlight individual's vulnerability for kidney selling. In another example, global transplant medicine (top left) and dysregulation therein is contributed by policy voids/loopholes and is further added by physician's attitude towards kidney patients and lackadaisical attitude to intervene against highly corrupt medical system. Development of this causal loop diagram (CLD) is based on an interactive process of literature review incorporating author's normative vision added by peer feedback and consultations. Combining literature reviews and authors' and experts' reflection complemented the analysis process. For further information in CLDs, please follow the online resource from the John Hopkins University (https://www.coursera.org/learn/systems-thinking).

number of data coders [40], coding tree [34] and feedback from the respondents in the findings [36].

## Thematic analysis

A thematic analysis utilizing the lens of power relations and social inequities has been used to conduct this review [27]. Most of the included studies have assessed the reasons of the commercial kidney sellers at an individual and societal level. However, the intermediate and macro level assessment were limited. The main themes identified in the analysis are presented in **S3 Table** along with participant quotes and interpretation from the author from contributing studies (Fig 3).

**Individual level.** At the individual level, poverty was the major reason for kidney selling, nonetheless, there were other reasons such as lack of information, desperation to alleviate poverty, debt, buying materials after kidney selling such as motorbikes, telephone and extra piece of land, desire for financial freedom, and altruism [2, 37, 39].

Many participants sold their kidneys due to lack of better economic conditions, such as jobs, basic facilities example, houses, and education for their children, as well as to repay their debt. It was carried out to sustain the basic family conditions. Some kidney sellers were vulnerable to sell their kidney because of their state of poverty, while others tended to broker kidney selling in the community to alleviate their level of poverty as well as to foster their economic condition. The study in Indian sub-continent showed poverty as the major contributor for

kidney selling [2]. Participants described selling their kidneys as a way to earn a living, manage household income, and alleviate poverty. The studies from India and Pakistan showed the kidney selling to relieve their debt and escape away from the debt trap [5, 25]. However, none of the participants were able to achieve financial freedom. Conversely, they were rather drawn into additional debt cycle and had poor health conditions which further restricted them to earn a living [2, 5, 25, 36].

Participants also shared their willingness to donate kidneys; however, they were compensated with financial deals and other benefits such as child education. Some participants claimed that their deeds were just altruistic rooted in religious beliefs [22, 41].

Some of the studies have reported the discordance in respondents' reasons for kidney selling. Even if respondents stated that they donated their kidneys to their relatives, and friends because of altruism, their inner motivations were conflicted with financial gains to compensate for employment, insecurity, and pressure to settle in urban areas which were found to be the major reasons for being drawn into kidney selling [42].

In a study conducted in the Philippines, participants compared the financial gains through kidney selling against the salary scales from jobs and also compared with how much money were made by other kidney sellers. Kidney sellers found justifications for kidney selling through such comparisons [35, 39].

**Societal level.** At the societal level, the studies also reported a broad communal trend of emulating each other rooted in social conformism. Conformism has been understood as a social dynamic reflecting the social cohesiveness, nonetheless, its role and impacts are also affected at an individual level. For instance, a person may act against the social norms of cohesiveness and conformity. The research carried out in Nepal highlighted the role of brokerage, while the study conducted in India highlighted the significance of social and family duties that contributed to gender roles. Furthermore, the investigations conducted in Pakistan and the Philippines emphasized the role of family obligations [2, 3, 37, 38].

Some of the studies from India showed that the male participants shared their kidney selling as their duty as they were the head of the household to fetch needs for the family. At the same time, females sold their kidneys to spare their husbands as they believed that the removal of kidney and surgical procedure could compromise their male partner's ability to work. Women were also convinced to sell the kidneys as they were persuaded that they did not have to undertake physical responsibilities [2, 3].

Some of the participants were pressurized with the social responsibility of dowry. Indian society expected dowry during their daughter's wedding, and males sold their kidneys to raise the funds for their daughter's wedding. Some participants sold their kidneys to fulfill the needs of the family; as there was no social security, they had to maintain the requirements of the family [2].

Some of the participants sold their kidneys due to ignorance and unawareness regarding the function of kidneys. Brokers had special role in persuading and convincing community members to sell their kidneys by manipulating the information related to kidney and its role. For instance, brokers misinformed the potential sellers that two kidneys were unnecessary, and one kidney could perform all the required function. Brokers also offered examples of their neighbors on how they had sold their kidneys in the past and how their health has remained intact after selling the kidneys. Few brokers also lured the poor community members for the employment purpose and persuaded for the better future. Because of ignorance often in remote regions of Nepal, community members were found to be gullible leading to falling victim to the brokers [38].

**Intermediate and macro level.** At the intermediate and macro level, a few studies (India and the Philippines) reported how cities with advanced medical facilities functioned as facilitators and meeting point for rich and poor for kidney trade [2]. Other studies reported the lack of social

welfare and the stringent policy from the government to protect the vulnerable people from selling their kidneys. However, there were no specific quotes presented for the discussion at intermediate and macro level [2, 5, 31, 34]. Study from Nepal reported the policy loopholes, the role of brokers, and unethical practices by the medical personnel as the main reasons for kidney selling [38].

## Discussion

This review elucidates a multitude of factors that contribute to the selling of kidneys, which are entrenched at the individual, societal, and structural levels. Some of the salient drivers behind kidney selling encompassed impoverishment, inadequate opportunities for livelihood, subjugation by brokers, lack of awareness, deceit, familial obligations, poorly regulated medical practices, including advanced biotechnology.

In this study, qualitative evidence is synthesized using a CMA approach to explore the reasons for kidney selling. The origin of kidney selling are discussed in the literature extensively, and are often attributed to the social conditions of the individuals [2, 5, 24–26, 36, 38]. Kidney selling has also been the result of short-term, impulsive decision making such as to buy ostentatious motorbikes and mobile phones in Nepal [26]. The selling of body parts to exchange with the ostentatious goods is perhaps an extreme form of transaction. The transaction of organs, blood to fulfill the daily needs are the grave situation [43]. Kidney selling was a source of earning money and obliged people to demonize poverty as a curse; hence many respondents in the articles attempted to sell their body and relieve their debt and overcome their poverty. Our review corroborates the findings of previous studies, which have identified economic factors as the primary drivers of kidney vending in India, Pakistan, Bangladesh, and Nepal [24–26, 39, 42]. Furthermore, some of the studies also described vulnerability to alleviate poverty as the reason for sales [5, 31]. Breaking the debt cycle by selling kidney was other main compensatory mechanisms among poor people [2].

Context as a facilitator was highlighted by Cohen et al that reported how Chennai, India for instance was a confluence for rich and poor with advanced medical facilities to support kidney transplantation [2]. Chennai also was an example for broader structural contributor for kidney selling. The convergence of wealthy and urban impoverished individuals comprised areas where the poor individuals experienced comparable deficits [2]. As a compensation to poverty, the poor people felt motivated to sell their kidneys which was facilitated by Chennai's advanced medical infrastructure for kidney transplantation. Increasing urbanization with poverty pockets offers a unique context for syndemics, that can foster both poverty associated diseases and life-style related diseases [21].

Study from Nepal and India identified cultural phenomena as the main reason for the kidney selling [2, 38]. The social conformism was found to be the main reasons for kidney selling in rural and remote regions in India, Nepal, and Pakistan [2, 5, 38]. In the identical research, familial gender roles were identified as the impetus for vended organs. Specifically, females were coerced into selling their kidneys by highlighting their domestic responsibilities, in contrast to males' external employment. The enactment of customary practices like dowry in the context of a daughter's nuptials, as well as the assumption of familial obligations, continue to exert significant societal influence among the populace of the Indian subcontinent. It is commonly acknowledged that such societal pressures are the main driving force behind the selling of kidneys [2, 38]. A study from Nepal also stated that the family responsibility their roles within the household as a reason for kidney selling [26, 38]. Similarly, gendered views on the role of kidney played critical role in kidney selling. In LMICs, a lot of High-Income Countries (HIC) patients approach for cheap kidneys. A study from Nigeria showed transplant tourism as an enabling factor for the kidney sales [34, 44].

At a macro level, the proliferation of biotechnology within LMICs has been identified as a significant factor contributing to the phenomenon of kidney selling. The rise of biomedicine has resulted in overuse and misuse of technology as is evident among the sellers who are unaware about the nature of kidney transplant and buyers who exploit the kidney transplant situation [34]. Furthermore, the dominance of biomedicine in responding to the healthcare requirements of individuals has resulted in unanticipated consequences for the people [34]. The increasing techno-ignorance among sub-population may have also contributed to increased submission to advanced medical procedures including the exploitations.

Individuals with end-stage renal diseases, who are genuinely in need of kidney transplants, are being deprived of easy access to donors, thereby necessitating the involvement of brokers and resulting in needless haggling and negotiations [10]. The concept of intermediation is of significant importance and has been duly noted in the scenario. Brokers often deceived donors from LMICs and negotiated with the recipients serving their commercial interests. Few studies also reported lack of adequate information, social conformism, gullibility rooted to ignorance as major contributors of kidney selling [2]. Furthermore, kidney selling was contributed chiefly by policy loopholes which were easily evadable, implying lack of good governance as the main reasons for kidney selling [38]. The easy documentation process (preparing false kinship) [45], open-border between India-Nepal, exploitation of documents required for bureaucratic purposes are easily forged and the policies are easily manipulated [35].

Although the reviewed studies primarily focused on the economic and cultural factors influencing kidney sales, few studies took into account the impact of policy and biotechnology. This suggests that while individual and societal factors play a role in kidney selling, other factors must also be considered. Therefore, a broader structural framework is necessary to fully understand the reasons behind kidney selling, particularly the policies and the international treaties regulating such policies across the countries that allow brokers to operate. Further research is needed to address this issue, especially given recent reports indicating that bio-technology and the global transplantation business are contributing to the growth of kidney trade [46]. Also, in recent years, kidney selling has multiplied globally because of the medical 'mafias' who have been able to exploit the policies systematically [47]. The mal-intentions of local brokers to facilitate the kidney selling has increased often expediated by the people who work at the higher echelons within the health care structure. For instance, health workers are often drawn into illegitimate kidney transplantation inadvertently mostly because of the completeness of medical documentations organized by brokers. In addition, the kidney sellers often naïve and under the circumstantial pressure stemming from their needs, thus can affect the authenticity of informed consent. An informed consent entails 1. providing adequate information by health care workers to participants; 2. participants understanding the offered information; and 3. participants are able to make a voluntary decision to proposed intervention. In case of kidney selling, participants may simply make decision under the influence of incentives (inducement), thus violating the ethical principles of informed consent [36, 48, 49]. The induced participation is exacerbated in scenarios where kidney sellers face structural barriers such as limited access to education, healthcare, legal assistance, and other relevant safeguards compromising the authentic informed consent [50]. Power differentials embedded in social inequalities also exacerbate the kidney selling whereby kidney sellers may feel coerced or pressured to consent without fully grasping the potential consequences and alternative options. Consequently, social inequality and power imbalances not only compromise the initial consent process but also have enduring effects on the well-being of kidney sellers [51]. Individuals from disadvantaged backgrounds often lack access to quality healthcare or post-operative support, heightening their susceptibility to complications and long-term health risks associated

with kidney donation [52]. Addressing these disparities is imperative to uphold ethical standards and ensure the welfare of all individuals involved in the kidney trade.

### Strengths and limitations

Use of CMA approach that entails analysis at individual, micro-social, intermediate and macro-social level highlighting the factors and their relationship affecting kidney selling is a comprehensive method. This review allowed us to understand the evidence at the individual and community level and revealed lack of research on policy, politics, governance, transnational policies and medical fraternity, ethical practices, and transnational laws through visualized prominent themes by geographical location (countries) which offers a distributive landscape of the kidney selling as opposed to only thematic presentations. Nonetheless, the impact of context and mechanisms underpinning specific phenomenon is beyond its scope and may require utilization of more theoretical approaches such as realist review. This review falls short in exploring ethical implications of kidney trafficking, such as how trafficking are imbued in poverty, inducement and vulnerability, thus calls for future studies exploring the ethical analysis around kidney selling and trafficking. Moreover, it is important to acknowledge limitations which relies on qualitative data means that it does not fully capture the quantitative aspects or the relative impact of various factors on kidney selling compared to others. These limitations underscore the importance of further research to address gaps in understanding and provide a more comprehensive analysis of the ethical dimensions of kidney trafficking.

### Conclusion

Despite wide spectrum of reasons to sell kidneys around the globe, there were paucities in efforts towards its mitigation. Kidney selling is a complex phenomenon and is shaped by myriad factors and their interactions. Factors and social rationales identified in this study warrant their interpretation aligning with the local social, cultural and political context and thus cannot be inferred in isolation.

### Supporting information

**S1 Checklist. PRISMA-ScR checklist.**
(PDF)

**S1 Table. Search words used.**
(DOCX)

**S2 Table. Comprehensiveness of reporting of included studies (COREQ).**
(DOCX)

**S3 Table. Summary of the included studies.**
(DOCX)

**S4 Table. Thematic findings from the studies.**
(DOCX)

### Author Contributions

**Conceptualization:** Bijaya Shrestha, Luechai Sringernyuang, Manash Shrestha, Bipin Adhikari.

**Data curation:** Bijaya Shrestha, Manash Shrestha, Binita Shrestha, Anuska Adhikari.

**Formal analysis:** Bijaya Shrestha, Manash Shrestha, Bipin Adhikari.

**Investigation:** Bijaya Shrestha, Manash Shrestha, Bipin Adhikari.

**Methodology:** Bijaya Shrestha, Luechai Sringernyuang, Manash Shrestha, Binita Shrestha, Anuska Adhikari, Bipin Adhikari.

**Project administration:** Bijaya Shrestha, Bipin Adhikari.

**Software:** Bijaya Shrestha, Manash Shrestha, Bipin Adhikari.

**Supervision:** Luechai Sringernyuang, Manash Shrestha, Bipin Adhikari.

**Validation:** Bijaya Shrestha, Binita Shrestha, Shiva Raj Mishra, Bipin Adhikari.

**Visualization:** Dev Ram Sunuwar, Shiva Raj Mishra.

**Writing – original draft:** Bijaya Shrestha, Manash Shrestha, Shiva Raj Mishra, Bipin Adhikari.

**Writing – review & editing:** Bijaya Shrestha, Luechai Sringernyuang, Manash Shrestha, Shiva Raj Mishra, Bipin Adhikari.

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
