## [Decision Letter · Decision Letter 0]

20 Nov 2023

PGPH-D-23-01387

Why do people sell their kidneys? A thematic synthesis of qualitative evidence

Dear Dr. Shrestha,

Thank you for submitting your manuscript to PLOS Global Public Health. After careful consideration, we feel that it has merit but does not fully meet PLOS Global Public Health’s publication criteria as it currently stands. Therefore, we invite you to submit a revised version of the manuscript that addresses the points raised during the review process.

A rebuttal letter that responds to each point raised by the editor and reviewer(s). You should upload this letter as a separate file labeled 'Response to Reviewers'.A marked-up copy of your manuscript that highlights changes made to the original version. You should upload this as a separate file labeled 'Revised Manuscript with Track Changes'.An unmarked version of your revised paper without tracked changes. You should upload this as a separate file labeled 'Manuscript'

We look forward to receiving your revised manuscript.

Kind regards,

Manish Barman, MD., MSc., FRCP

Academic Editor

Journal Requirements:

Additional Editor Comments (if provided):

Dear Authors

Indeed a thought provoking and a novel take on the subject.

I specifically waited a bit longer to receive an extensive and diverse feedback on your submission by our esteemed colleagues.

Kindly go through these valuable suggestions (mainly Lucas, Sharraf and Kathrin)

I am quite sure incorporating these suggestions in your work would definitely improve the paper impact.

Looking forward to your revised manuscript

Regards

Manish

Reviewers' comments:

Reviewer's Responses to Questions

**Comments to the Author**

1. Does this manuscript meet PLOS Global Public Health’s publication criteria? Is the manuscript technically sound, and do the data support the conclusions? The manuscript must describe methodologically and ethically rigorous research with conclusions that are appropriately drawn based on the data presented.

Reviewer #1: Yes

Reviewer #2: Yes

Reviewer #3: Yes

Reviewer #4: Yes

Reviewer #5: Yes

Reviewer #6: Yes

Reviewer #7: Yes

Reviewer #8: Yes

Reviewer #9: Yes

2. Has the statistical analysis been performed appropriately and rigorously?

Reviewer #1: N/A

Reviewer #2: Yes

Reviewer #3: N/A

Reviewer #4: N/A

Reviewer #5: N/A

Reviewer #6: Yes

Reviewer #7: Yes

Reviewer #8: N/A

Reviewer #9: Yes

3. Have the authors made all data underlying the findings in their manuscript fully available (please refer to the Data Availability Statement at the start of the manuscript PDF file)?

Reviewer #1: Yes

Reviewer #2: Yes

Reviewer #3: Yes

Reviewer #4: Yes

Reviewer #5: Yes

Reviewer #6: Yes

Reviewer #7: Yes

Reviewer #8: Yes

Reviewer #9: Yes

4. Is the manuscript presented in an intelligible fashion and written in standard English?

Reviewer #1: Yes

Reviewer #2: Yes

Reviewer #3: Yes

Reviewer #4: Yes

Reviewer #5: Yes

Reviewer #6: Yes

Reviewer #7: Yes

Reviewer #8: Yes

Reviewer #9: Yes

5. Review Comments to the Author

Reviewer #1: Title and abstract:

The title effectively conveys the paper's focus on kidney selling and aligns well with the paper's content, adhering to best practices for research paper titles. The abstract provides a clear and concise summary of the paper, offering a good overview of the paper's content in a structured manner.

Introduction:

Strengths:

The introduction effectively highlights the global significance of kidney selling, its supply and demand dynamics, and the role of policies and medical perspectives involving this phenomenon. Besides providing the context by emphasizing the importance of the issue of kidney selling, it clearly outlines the paper's research objectives and scope.

Areas for Improvement:

However, the introduction lacks emphasis on the ethical implications of this practice, whether legal or illegal. Addressing ethical concerns is vital in understanding the broader implications of kidney selling and its impact on individuals and communities.

Materials and Methods:

Strengths:

The manuscript exhibits a good methodological rigor in its Materials and Methods section. It adheres to ENTREQ guidelines, registers on PROSPERO for transparency, and clearly specifies databases and the search strategy. The delineation of publication dates is well-justified, and the inclusion and exclusion criteria, along with the study selection process, are clearly articulated. The use of COREQ for quality assessment reflects a systematic approach. Furthermore, the incorporation of critical medical anthropology (CMA) provides a valuable theoretical lens to understand the socio-cultural and economic dynamics of kidney selling. Overall, most of the methodological choices adopted seem adequate for the purpose of this study.

Areas for Improvement:

1. The section would benefit from a brief explanation of the reasons for choosing the specific databases and search terms.

2. There is a mismatch between the keywords provided in the body text and the supplementary table 4. For example, “commerce” (not “deal”) is included in the PubMed search, yet not included in the other databases search. If there is a reason for this choice, this should be mentioned in the manuscript.

3. It is not clear how the Google Scholar search was conducted. This should be better detailed.

4. There is a duplicated word (“studies”) on line 127 that should be corrected.

5. While the description of data extraction and quality assessment is clear, more detail regarding potential discrepancies in study classification or quality assessment between the two reviewers should be provided. Elaboration on how these discrepancies were resolved would enhance transparency and robustness.

6. On supplementary table 1, the reporting lacks uniformity. For instance, for most studies, the authors mention only countries, while in study 2, they also included cities. The reporting of dates should also follow a pattern.

7. The manuscript should adopt same wording for core concepts, such as “social inequality” and power differentials (line 107) or power disparities (line 166).

8. Moreover, the manuscript would gain clarity if it clearly delimitates the core concepts adopted by this study, particularly expressing which definitions were adopted and why.

Results:

Strengths:

The Results section is well-organized, presenting findings from the selected studies in a clear and structured manner. The section effectively summarizes the key themes and factors influencing kidney selling based on the selected studies. Findings are supported by quotes from participants, enhancing the credibility of the results. Geographic variability is clearly highlighted though the map, providing a facilitated view of the issue on a global scale. The use of a causal loop diagram (CLD) in Figure 3 offers a visual representation of the complex factors influencing kidney selling and how they are interrelated, enhancing the clarity of the results.

Areas for Improvement:

1. Providing a brief summary or introductory paragraph at the beginning of the "Results" section could offer readers an overview of the key findings and their significance.

2. Consider relocating the description of CLD development to the "Materials and Methods" section to improve the flow and alignment with the study's methodology.

3. The manuscript would benefit from a cohesive visual identity. For instance, using a consistent color palette for diagrams and figures could enhance the overall presentation. The CLD currently employs contrasting colors and a small font size, which might impede readers in easily grasping the intended message.

4. To enhance open science practices in this manuscript, the authors might consider providing access to their literature records database stored in EndNote. This would improve transparency and support validation, replication, reanalysis, and study reproducibility.

5. Please note that after page 8, the formatting of page numbers and line numbers become inconsistent. Consider correcting it to facilitate further reviews.

Discussion and Conclusion:

Strengths:

The "Discussion" section effectively draw upon the study results to analyze the multifaceted factors influencing kidney selling, providing a comprehensive understanding of this complex issue. The discussion includes relevant contextual information about kidney selling, particularly within low- and middle-income countries, and adequately emphasizes the need for a more comprehensive approach to understanding and addressing this phenomenon. The conclusion provides a concise summary, effectively highlighting the complex nature of kidney selling.

Areas for Improvement:

1. Consider aligning the discussion more clearly with the concepts presented in the research question in the introduction section. While the authors addressed the aspect of "social inequality" in kidney selling, the discussion of "power differentials" appears to be less prominent or not clearly articulated. Ensuring a stronger link between these concepts can strengthen the overall coherence of the analysis and guarantee better robustness to the study.

2. While not the main research question, the "Discussion" section could benefit from greater emphasis on the ethical implications of kidney selling. Exploring ethical aspects, such as exploitation, informed consent, and the well-being of kidney sellers, would provide a more comprehensive analysis, especially regarding how these ethical considerations relate to "social inequality" and "power differentials," which are the central research focus.

3. To enhance the discussion, consider examining the study results while taking into account the inherent biases in the analyzed studies. This approach would add depth to the interpretation of findings and acknowledge potential limitations.

Final consideration: to ensure transparency to this process, please note that I used ChatGPT-3.5 for proofreading my review, but I am the sole author of the content of this peer review.

Reviewer #2: The tools and materials used in the data search, extraction, synthesis and analysis are quite relevant and of good standard that gives a clear picture of the designs, analysis, and findings from the literatures that were explored. Categorization of the analysis into geographical variation was also a very good approach as this was able to clearly point out different perspectives, based on different societal context, of why people sell their kidneys. Hence, I recommended this paper to be published.

Reviewer #3: I greatly enjoyed reading the manuscript, which sheds light on a relevant public health issue that requires being properly addressed by authorities worldwide. Appendices are very rich in information. I am attaching detailed review in PDF.

Reviewer #4: I appreciate the effort made by the authors team.

Author had followed all standard scientific methods and study submitted inline with the prospero protocol. However, the topic looks very simple and repetation of well known research question.

Some suggestion,

1.Introduction need little more clarity and citation in line number 57 & 72

2. Author had failed to establish the concepts related to power disparities and social inequalities, political and sociocultural aspects role in kidney selling-Introduction

3.Concept of stigma was unclear-63 line number

4. Repeated word line-127

5. What is the difference between line number 205-207 & 211-213

6.Supplementary-3, is not appropriately reflected in result section

Reviewer #5: Major issues

1) The introduction lacks a clear motivation for the topic, a review of the literature, and the added value of this particular paper. Furthermore, the introduction is sometimes repetitive and does not follow a clear line of argumentation, but often drifts into background description. I.e.

a. It is not clear how big the issue is; how does it vary between countries? Further, it would also be useful to provide the reader with some figures.

b. It is not clear what the evidence is on legal and illegal kidney selling; how do they differ? Furthermore, what is the difference between illegal kidney selling with and without consent?

c. As the paper states that the majority of transplants in LMICs come from illegal trafficking, it would be beneficial to have more emphasis on how this was done.

d. The first and second paragraphs are repetitive. Overall, the introduction would benefit from a clearer line of argumentation.

e. The background description of e.g., transplantation techniques, health consequences, and legal framework should be kept to a minimum; but should be expanded in an additional background/context section.

f. It is not very clear what the contribution of the paper is. The following two sentences in the last paragraph of the introduction are confusing because it is not explained why this is the scope in particular: “In this review, we use a critical approach to explore social inequality and power differentials as the primary contributors of health and health care. This review also analyzes the relationship between health status at an individual and societal level (16).”

2) The paper would benefit from a clearer explanation of study selection decisions and a discussion of the consequences of selection:

a. Only studies published after 1987 are included in the sample, because the WHO declared organ selling illegal in 1987. The authors should make this clear in the introduction and explain how this affects their results.

b. The selection of studies excludes three quantitative studies because they did not provide new evidence. As these studies are based on a survey of kidney sellers, the authors' argument does not seem obvious. The authors should explain in more detail why they did not provide new evidence and why they cannot contribute to the results.

3) It is not clear which results are based on how many articles and from which countries. It would be beneficial to provide more detailed information. I.e.,

a. In the list of major themes of the articles, it would be beneficial to indicate how many and which countries address these themes. Furthermore, sometimes country examples are mentioned (e.g., themes 1. And 6.) and sometimes not, which is not transparent for the reader.

b. In the discussion of the individual level and societal levels, mainly studies from the Asia continent are mentioned. However, Figure 1, shows that studies from South America, Europe and Africa are also evaluated. It would be beneficial to have a transparent discussion of the results to avoid a bias of over-reporting Asian results.

c. It is not clear to the reader why Figure 3 only looks at factors influencing kidney selling in Nepal. How would it be different in other countries based on the reviewed papers? Additionally, the figure is not embedded in the text. It is highly recommended to explain how this figure was generated (based on what data), what it can show, and what its limitations are.

4) The overall interpretation and reporting of the results throughout the paper is often not inconsistent and lacks explanation. They should be coherent, well structured and well explained. I.e.,

a. The abstract states that the paper shows various similarities across different geographical regions, but the authors only mention countries on the Asian continent rather than comparing them with the other American, European and African countries.

b. The results in the abstract do not summarise well the overall results of the paper, e.g. they do not give insights into the meso and micro level, they mention illiteracy as a micro factor which is not mentioned at all in the rest of the paper.

c. Usually, reference is made to the individual and societal levels. However, the discussion also refers to the structural level, which was not discussed before.

d. As concluded in the "Thematic Analysis" section, the assessment of the intermediate and macro levels was limited. I would therefore recommend that it be stated less prominently that the analysis provides insight into the intermediate and macro-social levels.

e. It would be more transparent if the authors also discussed the limitation of the sample in the sense that only a few countries were observed and that the mainly Asian countries were mentioned in the results and discussion section.

Minor issues

1) The figures should be in the order in which they are presented in the text. They should also be made more readable.

2) It would be useful for the reader to see in Supplementary Table 1 the number of observations/interviews for each of the studies.

3) There are several typographical errors.

Reviewer #6: Overall, the paper is well-written and easy to follow, and the aims are well-formulated and clearly communicated. However, I have some concerns about some sections as well as some other less substantial questions and comments. My comments are ordered in two sections below: Major comments and minor comments.

The major comments are specifically for the method and result sections which should be reconsidered. Although studies from all over the world have been included in the systematic review, most of the discussion/explanations are based on the Asian countries only.

Also, there are lots of formatting issues which should be corrected.

Reviewer #7: This is a very interesting systematic review (qualitative) in the field of public health. It describes why kidney trade occurs and how it is also affecting the health of most people in developing countries. The manuscript is well-written showing empirical evidence.

Authors should address these attached minor issues before the editor-in-chief will accept this manuscript. The Reviewer's comments are attached.

Reviewer #8: The paper is very interesting and has wonderful insights. It is well written clear and easy to understand. The methodological approach is very well explained and is easily replicable in an objective manner.

Reviewer #9: Overall Comment: The authors have systematically reviewed published qualitative studies on motivators and facilitators of Kidney selling globally. This important study synthesizes the factors that aggravate kidney selling and has the potential for policy reform. I have provided a few comments. The authors may want to consider them before publication.

Abstract: In the Result section, micro-level factors were presented only; however, the Conclusion section includes meso, & macro-level factors. I suggest including some meso and macro factors in the Results of the Abstract as well.

Introduction: The Last sentence of 1st paragraph (Line no. 61-63, Page.3) fits better in line no. 93 (Page.4).

Materials and Methods: The authors should explain more about ENTREQ (Line no.111, Page.4) as they have done for the COREQ. It is unclear how those two were utilized in this study, especially how ENTREQ differed from COREQ.

Data synthesis and analysis: The sentence starting with "It critically examines the ways in which power, ..........."(Line no. 77-79 in Page.6 & Line no. 180-181 in Page.7) doesn't fit much with the research topic. The topic is Kidney selling, but this paragraph only talks about healthcare access and policy things. So, I suggest to edit this sentence.

Results: In the 7th major theme, the word lax policies is not clear to me. What is it? Just check whether this is a typo error.

Results: Thematic analysis: The 1st sentence (Line no.16-17, Page.2; Don't confuse, the line no. & page no. are not sequential. So, check line no. of Result section) is about methods which was already reported in the Methods section. Hence, for me, it is not necessary to rewrite here. After the last sentence, authors need to add some text to connect with the following subheadings so that readers will understand that "Individual level" is the subheading of results from the Thematic analysis.

Results: Societal Level: Capitalization of 'L' in level is not consistent with Individual level. Please make uniform capitalization.

In Figure 3, the description of the photo is long. I am confused about whether the paragraph is an interpretation or description of the photo. It looks like the authors wrote everything in the caption. If the paragraph is about interpretation, please write in a paragraph rather than a caption.

Reference: The short form of the journal name is inconsistent. Journal names in some references were shortened, and the majority were kept as it is. I suggest authors make a consistent referencing style. Furthermore, the Nepali name in reference no.43 should be standardized. Please check how to cite references in a foreign language in the PLOSGP style and correct it.

Thank you.

6. PLOS authors have the option to publish the peer review history of their article (what does this mean?). If published, this will include your full peer review and any attached files.

**Do you want your identity to be public for this peer review?** For information about this choice, including consent withdrawal, please see our Privacy Policy.

Reviewer #1: No

Reviewer #2: **Yes: **Buba Darboe

Reviewer #3: No

Reviewer #4: No

Reviewer #5: No

Reviewer #6: No

Reviewer #7: No

Reviewer #8: No

Reviewer #9: **Yes: **Krishna Prasad Sapkota

---

## [Decision Letter · Decision Letter 1]

22 Feb 2024

Why do people sell their kidneys? A thematic synthesis of qualitative evidence

PGPH-D-23-01387R1

Dear Mr Shrestha,

We are pleased to inform you that your manuscript 'Why do people sell their kidneys? A thematic synthesis of qualitative evidence' has been provisionally accepted for publication in PLOS Global Public Health.

Best regards,

Manish Barman, MD., MSc., FRCP

Academic Editor

After extensively going through the reviewers’ comments, I am of the opinion that the authors have now incorporated the much-required changes/ suggestions in their revised manuscript. Authors have duly followed the guidelines of ENTREQ for this qualitative systematic review and the protocol is duly registered in the PROSPERO database (CRD42020197541). Google scholar search has been done additionally to not miss any important grey literature. Choosing search terms and important databases selection is adequate and there is no further need for any explanations by the authors. Data extraction and the quality is explicitly clear and does not need to be addressed any further.

Further recommendations made by one of the reviewer using terms as - establishing stronger links-" greater emphasis"- are quite subjective terms and difficult to be incorporated by any writer/ author. The authors have emphasized their observations with reasonable evidence. In my view the authors have adequately addressed all the major concerns and meticulously tabulated the changes for clarity.

The revised script has been recommended as ACCEPT by four reviewers except one.

I would also recommend acceptance and publication of the manuscript.

Reviewer Comments (if any, and for reference):

Reviewer's Responses to Questions

**Comments to the Author**

1. If the authors have adequately addressed your comments raised in a previous round of review and you feel that this manuscript is now acceptable for publication, you may indicate that here to bypass the “Comments to the Author” section, enter your conflict of interest statement in the “Confidential to Editor” section, and submit your "Accept" recommendation.

Reviewer #1: (No Response)

Reviewer #2: All comments have been addressed

Reviewer #7: All comments have been addressed

Reviewer #8: All comments have been addressed

Reviewer #9: All comments have been addressed

2. Does this manuscript meet PLOS Global Public Health’s publication criteria? Is the manuscript technically sound, and do the data support the conclusions? The manuscript must describe methodologically and ethically rigorous research with conclusions that are appropriately drawn based on the data presented.

Reviewer #1: Yes

Reviewer #2: Yes

Reviewer #7: Yes

Reviewer #8: (No Response)

Reviewer #9: Yes

3. Has the statistical analysis been performed appropriately and rigorously?

Reviewer #1: N/A

Reviewer #2: Yes

Reviewer #7: Yes

Reviewer #8: (No Response)

Reviewer #9: Yes

4. Have the authors made all data underlying the findings in their manuscript fully available (please refer to the Data Availability Statement at the start of the manuscript PDF file)?

Reviewer #1: Yes

Reviewer #2: Yes

Reviewer #7: Yes

Reviewer #8: (No Response)

Reviewer #9: Yes

5. Is the manuscript presented in an intelligible fashion and written in standard English?

Reviewer #1: Yes

Reviewer #2: Yes

Reviewer #7: Yes

Reviewer #8: (No Response)

Reviewer #9: Yes

6. Review Comments to the Author

Reviewer #1: Dear authors,

I am glad that you decided to continue the improvement of your article since this is a very important topic and you conducted a commendable research study that should indeed be published. I noticed an improvement based on other reviewer's comments and I appreciate it. Unfortunately, I have not noticed you addressing my comments from the previous round. I understand that you may disagree with my comments, but they need to be addressed so you can also give me the opportunity to reflect on my decision and comments. Therefore, I copy and paste my comments from the previous round below and I highlight the ones which were already corrected by you and the ones that were not yet addressed.

Introduction: the introduction lacks emphasis on the ethical implications of this practice, whether legal or illegal. Addressing ethical concerns is vital in understanding the broader implications of kidney selling and its impact on individuals and communities. (nicely included in paragraph lines 91-110).

Materials and Methods:

1. The section would benefit from a brief explanation of the reasons for choosing the specific databases and search terms. (NOT addressed.)

2. There is a mismatch between the keywords provided in the body text and the supplementary table 4. For example, “commerce” (not “deal”) is included in the PubMed search, yet not included in the other databases search. If there is a reason for this choice, this should be mentioned in the manuscript. (NOT addressed.)

3. It is not clear how the Google Scholar search was conducted. This should be better detailed. (NOT addressed.)

4. There is a duplicated word (“studies”) on line 127 that should be corrected. (OK, corrected.)

5. While the description of data extraction and quality assessment is clear, more detail regarding potential discrepancies in study classification or quality assessment between the two reviewers should be provided. Elaboration on how these discrepancies were resolved would enhance transparency and robustness. (NOT addressed.)

6. On supplementary table 1, the reporting lacks uniformity. For instance, for most studies, the authors mention only countries, while in study 2, they also included cities. The reporting of dates should also follow a pattern. (NOT addressed. Also, something changed and now there is a conflict of naming, e.g., supplementary table 3.docx is named Supplementary table 1 within the file text.)

7. The manuscript should adopt same wording for core concepts, such as “social inequality” and power differentials (line 107) or power disparities (line 166). (NOT addressed.)

8. Moreover, the manuscript would gain clarity if it clearly delimitates the core concepts adopted by this study, particularly expressing which definitions were adopted and why. (NOT addressed.

Discussion and Conclusion:

1. Consider aligning the discussion more clearly with the concepts presented in the research question in the introduction section. While the authors addressed the aspect of "social inequality" in kidney selling, the discussion of "power differentials" appears to be less prominent or not clearly articulated. Ensuring a stronger link between these concepts can strengthen the overall coherence of the analysis and guarantee better robustness to the study. (NOT addressed.)

2. While not the main research question, the "Discussion" section could benefit from greater emphasis on the ethical implications of kidney selling. Exploring ethical aspects, such as exploitation, informed consent, and the well-being of kidney sellers, would provide a more comprehensive analysis, especially regarding how these ethical considerations relate to "social inequality" and "power differentials," which are the central research focus. (NOT addressed. Included in the introduction, but not recalled for reflection in the Discussion section.)

3. To enhance the discussion, consider examining the study results while taking into account the inherent biases in the analyzed studies. This approach would add depth to the interpretation of findings and acknowledge potential limitations. (NOT addressed.)

Reviewer #2: (No Response)

Reviewer #7: The authors have appropriately addressed all the comments.

Reviewer #8: All the comments have been addressed. I have no other issues with the revised version of the paper.

Reviewer #9: Majority of issues were addressed by authors. Thank you very much, and good luck with the publication.

7. PLOS authors have the option to publish the peer review history of their article (what does this mean?). If published, this will include your full peer review and any attached files.

**Do you want your identity to be public for this peer review?** For information about this choice, including consent withdrawal, please see our Privacy Policy.

Reviewer #1: No

Reviewer #2: **Yes: **Buba Darboe

Reviewer #7: **Yes: **Silas Adjei-Gyamfi

Reviewer #8: No

Reviewer #9: No
